# Integrative Physiological and Transcriptome Analysis Reveals the Mechanism for the Repair of Sub-Lethally Injured *Escherichia coli* O157:H7 Induced by High Hydrostatic Pressure

**DOI:** 10.3390/foods11152377

**Published:** 2022-08-08

**Authors:** Jing-Yi Hao, Yu-Qing Lei, Jun-Yan Shi, Wan-Bin Zhao, Zhi-Lin Gan, Xin Hu, Ai-Dong Sun

**Affiliations:** 1College of Biological Sciences and Biotechnology, Beijing Forestry University, Beijing 100083, China; 2Beijing Key Laboratory of Forest Food Processing and Safety, Beijing 100083, China

**Keywords:** *Escherichia coli* O157:H7, high hydrostatic pressure, sub-lethally injury, repair mechanism, transcriptome analysis

## Abstract

The application of high hydrostatic pressure (HHP) technology in the food industry has generated potential safety hazards due to sub-lethally injured (SI) pathogenic bacteria in food products. To address these problems, this study explored the repair mechanisms of HHP-induced SI *Escherichia coli* O157:H7. First, the repair state of SI *E. coli* O157:H7 (400 MPa for 5 min) was identified, which was cultured for 2 h (37 °C) in a tryptose soya broth culture medium. We found that the intracellular protein content, adenosine triphosphate (ATP) content, and enzyme activities (superoxide dismutase, catalase, and ATPase) increased, and the morphology was repaired. The transcriptome was analyzed to investigate the molecular mechanisms of SI repair. Using cluster analysis, we identified 437 genes enriched in profile 1 (first down-regulated and then tending to be stable) and 731 genes in profile 2 (up-regulated after an initial down-regulation). KEGG analysis revealed that genes involved in cell membrane biosynthesis, oxidative phosphorylation, ribosome, and aminoacyl-tRNA biosynthesis pathways were enriched in profile 2, whereas cell-wall biosynthesis was enriched in profile 1. These findings provide insights into the repair process of SI *E. coli* O157:H7 induced by HHP.

## 1. Introduction

High hydrostatic pressure (HHP) is a promising non-thermal processing technique used for the inactivation of micro-organisms (bacteria, yeast, and mold) in food systems, because of the advantages of preserving nutritional and sensory characteristics [1,2]. HHP technology has been successfully commercialized in the past decade [3] in fruit and vegetable juices, meat, and dairy products. The global market for HHP foods reached approximately USD 9.8 billion in 2015 and is expected to attain a market value of USD 54.77 billion in 2025 [4].

*Escherichia coli* O157:H7 has been implicated in many food-borne outbreaks and posed a worldwide threat to public health [5,6]. According to the Centers for Disease Control and Prevention, beef and leafy vegetables were the sources of >25% of all reported *E. coli* outbreaks and >40% of related illnesses from 2003 to 2012 [7]. In addition, *E. coli* O157:H7 is highly virulent and causes inflammatory reactions, hemolytic uremic syndrome, and thrombocytopenic purpura [8], establishing this organism as a high-risk pollution challenge that must be addressed.

More dangerous still, sub-lethally injured (SI) cells (including SI *E. coli*) could be induced by HHP in various food matrixes, thereby limiting the promotion of HHP [9,10,11]. SI cells are induced by stresses that are not severe enough to destroy cells [12]. Studies found that typical food-borne pathogens (such as *Listeria innocua*, *E. coli*, and *Bacillus subtilis*) could be sub-lethally injured by HHP. The proposed mechanism could involve many targets, including reversible membrane injury, nucleic acid and proteins, and metabolic disorders. [13,14]. Injured cells could recover physiological function and virulence when they are repaired in favorable conditions, and then proliferate in food during storage, leading to potential health threats [15,16,17]. Though detection methods have been developed to detect SI cells, the results have proven inaccurate, leading to false negative results due to differences among the culture media [18].

Several studies focused on SI cells’ quantity, interfering factors, and mechanisms. Sokolowska et al. [19] found that 2.7 log of SI *E. coli* cells could be detected in phosphate-buffered saline (PBS) after 400 MPa treatment for 10 min with the increase in initial bacterial concentration, the content of SI cells increased. The SI processing could result in changes in morphology, protein denaturation, membrane damage, and oxidation stress [20,21]. Other studies focused on the repair conditions of SI cells. Yamamoto et al. [21] found that the repair of SI *E. coli*, *Listeria monocytogenes*, and *B. subtilis* induced by HHP (300–600 MPa) were related to the storage temperature and nutrient level of the repair medium, 25 °C and nutrient-rich medium were more suitable for the repair process. Shi et al. [12] found that the repair rate of SI *E. coli* O157:H7 induced by lactic acid could be increased by sodium pyruvate, Tween 80, or certain cations (Mn, Fe, or Zn) but not influenced by Mg or Ca. These findings suggest that the repair of SI cells is a common phenomenon with complex mechanisms. Unfortunately, few studies have focused on the mechanism of the repair process, especially from the molecular level. 

To fill this gap, we employed comparative transcriptome analysis for SI (induced by HHP) and repair of *E. coli* O157:H7 models to determine the differences in the transcriptional responses during the repair process. Physiological analysis was used to augment this analysis. Our research provided new insights into the underlying molecular and cellular mechanisms of the repair process, which would help to explore new ways for improving the HHP antibacterial effect and avoiding the repair of SI cells.

## 2. Materials and Methods

### 2.1. Strain and Culture Condition

We used *E. coli* O157:H7 NCTC 12900, preserved in Beijing Key Laboratory of Forest Food Processing and Safety. A single colony was inoculated into 50 mL of tryptose soya broth culture medium (TSB, Aoboxing Biotech Co., Ltd., Beijing, China) and then incubated at 37 °C and 180 rpm for 5 h (optical density at 600 nm = 0.86) in a shaker (TS-100B, Shanghai Tiancheng Experimental Instrument Manufacturing Co., Ltd., Shanghai, China) until the culture reached the medium segment of the logarithmic phase [22]. Bacterial pellets were harvested by centrifugation at 6000 rpm and 4 °C for 10 min and then washed three times with PBS (pH 7.2–7.4, Biotopped Co., Ltd., Beijing, China). The final bacterial suspension concentration was approximately 10^8^–10^9^ colony-forming units/mL.

### 2.2. Determination of Sub-Lethal Treatment Conditions 

The bacterial suspension was divided into sterile bags equally (10 mL per bag) and sealed tightly. The packed samples were soaked in deionized water in an airtight cabin, and then pressurized at 100, 200, 300, 400, and 500 MPa for 1, 3, or 5 min at 25 °C using HHP equipment (Shanghai Litu Ultra-High Voltage Equipment Co., Ltd., Shanghai, China). The constant pressure time did not include the pressure increase (2–3 min) or pressure-release (1–2 s) times. The treated bacterial suspension in identical conditions was coated on a non-selective medium (NS, tryptose soya agar medium, TSA) and a selective medium (SC, TSA with 3% NaCl) plates, respectively [22]. The plates were placed upside-down in an incubator for 24 h at 37 °C, and the sub-lethal effects were determined using the colony counting method. The SI rate, SI cell content, intact cell content, and dead cell content were calculated using the following formulas: A_control_, A_ns_, and A_sc_ present colony-forming units of untreated control, cells survived on the NS culture media, and cells survived on the SC media, respectively.
(1)$$\mathrm{SI}\;\mathrm{rate}\%=\frac{\left(A_{\mathrm{ns}}{\;-\;A}_{\mathrm{sc}}\right)}{A_{\mathrm{ns}}}\;\times \;100$$
(2)$$\mathrm{SI}\;\mathrm{cell}\;\mathrm{content}\%=\frac{{(A}_{\mathrm{ns}}\;{-\;A}_{\mathrm{sc}})}{A_{\mathrm{control}}}\;\times \;100$$
(3)$$\mathrm{Intact}\;\mathrm{cell}\;\mathrm{content}\%=\frac{A_{\mathrm{sc}}}{A_{\mathrm{control}}}\;\times \;100$$
(4)$$\mathrm{Dead}\;\mathrm{cell}\;\mathrm{content}\%=100\;-\;\frac{A_{\mathrm{ns}}}{A_{\mathrm{control}}}\;\times \;100$$

### 2.3. Determination of Repair Conditions

The SI *E. coli* O157:H7 was repaired in TSB, buffered peptone water (BPW), PBS, and minimal medium A (minA) medium, respectively; minA was prepared according to Hui et al. [17]. The repair method referred to Bi et al. [22]. Briefly, the SI *E. coli* O157:H7 induced by HHP treatment were centrifuged at 6000 rpm and 4 °C for 10 min to remove PBS. The bacterial pellets were re-suspended in equivalent media and the bacterial suspensions were incubated at 37 °C and 120 rpm in a shaker for 6 h. Then, 100 μL of bacterial solution diluted to the appropriate concentration was coated on NS and SC every hour, respectively. The plates were then placed in the incubator for 24 h at 37 °C, and the number of surviving colonies was recorded. The results were displayed in the form of the logarithm of the surviving colony number (lgS).

### 2.4. Determination of the Intracellular Protein Content and Adenosine Triphosphate (ATP) Contents

The intracellular protein content was determined using a total protein assay kit (BCA method, Beijing BioDee Biotechnology Co. Ltd., Beijing, China). Briefly, the bacterial suspensions subjected to various treatments were centrifuged at 6000 rpm and 4 °C for 10 min to obtain pellets. Each pellet was re-suspended in PBS and crushed using an ultrasonic processor (HY92-IIDN, Ningbo Scienta Biotechnology Co., Ltd., Ningbo, China) in an ice bath to obtain the intracellular solution. Then, the 20 μL intracellular solution was added to the 200 μL BCA solution and incubated at 37 °C for 1 h. The absorbance was determined at 562 nm to calculate the protein content using a standard curve (y = 0.0009x + 0.0068, R^2^ = 0.9993). The ATP content was determined using an assay kit (Jian Cheng Bioengineering Institute, Nanjing, China). To avoid the influence of phosphorus in PBS, the bacterial pellet was suspended in boiled sterile water for the subsequent ultrasonication, as described above. The intracellular solution was then boiled in water for 15 min and extracted for two minutes to perform the subsequent experiment according to the manufacturer’s instructions.

### 2.5. Observation of the Morphological Changes

Atomic force microscopy (AFM) was performed to observe the morphological changes of *E. coli* O157:H7 after various treatments. Bacterial suspensions of the untreated (UT), HHP, and repair groups (repaired for 4 and 8 h) were prepared, as described in Section 2.2 and Section 2.3, and fixed with glutaraldehyde (BioDee Biotechnology Co., Ltd., Beijing, China) overnight at 4 °C. Each mixture was washed three times with sterilized water and re-suspended in sterilized water. Each sample was dropped onto the mica sheet and naturally dried before observing using a Bruker Multimode 8 AFM (Bruker Corporation, Karlsruhe, Germany) in auto scan mode. Then, 20 areas of 509.6 × 509.6 nm^2^ were randomly selected in each treatment to count the root mean square roughness using NanoScope Analysis (version 1.40, Bruker Corporation, Karlsruhe, Germany).

### 2.6. Measurement of Enzyme Activities

Using a kit, a UV-vis spectrophotometer determined the catalase (CAT) activity (Jian Cheng Bioengineering Institute, Nanjing, China). Briefly, the equivalent intracellular solution was added into the testing tube and contrast tube before and after the addition of the stop solution, respectively. After standing at room temperature for 2 min, the absorbance was measured at 405 nm. The superoxide dismutase (SOD) activity was determined using a WST-1 method kit and a microplate reader (Multiskan FC, Thermo Fisher Scientific, Waltham, MA, USA). The intracellular solution was added to the testing well, and equivalent PBS was added to the contrast well instead of the intracellular solution. The reaction system was incubated at 37 °C for 30 min and then measured at 450 nm. The ATPase activity was measured using a minim ATP enzyme test kit (Jian Cheng Bioengineering Institute, Nanjing, China). To avoid the influence of phosphorus in PBS, the bacterial pellet was suspended in cold sterile water, and then ultrasonicated to obtain the enzyme solution. The results were measured at 636 nm using the microplate reader.

### 2.7. RNA Isolation and Library Construction

The *E. coli* O157:H7 UT group, the SI (HHP group, treated at 400 MPa for 5 min), and those repaired for 2 h in TSB (repair group) were quickly frozen in liquid nitrogen for the subsequent transcriptomic analysis. All groups were cultured in triplicate. Total RNA was extracted using a TTIzol reagent kit (Invitrogen, Carlsbad, CA, USA). RNA quality was tested on an Agilent 2100 Bioanalyzer (Agilent Technologies, Palo Alto, CA, USA) and checked using RNase free agarose gel electrophoresis. After the extraction, mRNA was enriched by removing rRNA by Ribo-Zero^TM^ magnetic kit (Epicentre, Madison, WI, USA). Finally, the ligation products were size selected by agarose gel electrophoresis, PCR amplified and sequenced using Illumina HiSeq2500 by Gene Denovo Biotechnology Co. (Guangzhou, China).

### 2.8. Transcriptomic Analysis 

For each transcription region, a fragment per kilobase of transcript per million mapped reads (FPKM) value was calculated. The false discovery rate (FDR) was calculated to correct for the multiple testing to adjust the *p*-value threshold. RNAs differential expression analysis was performed using DESeq2 software between different groups (and by edgeR between two samples). The genes with the parameter of FDR below 0.05 and absolute fold change (FC) ≥ 2 (or |log_2_FC| ≥ 1) were considered differential expression genes (DEGs). Gene ontology (GO) was performed for the DEGs for each treatment stage by mapping all DEGs to GO terms in the GO database. GO terms where *p*-value ≤ 0.05 were defined as significantly enriched GO terms in DEGs. Kyoto Encyclopedia of Genes and Genomes (KEGG, a public pathway-related database) was used to identify significantly enriched metabolic pathways and signal transduction pathways in DEGs. Pathways meeting the condition that *p*-value ≤ 0.05 were defined as significantly enriched pathways in DEGs. The transcriptome sequence of *E. coli* O157:H7 is deposited at the National Center of Biotechnology Information (NCBI) under the Bioproject number PRJNA769820.

### 2.9. Real Time Quantitative PCR (qRT-PCR) Validation

The qRT-PCR assay was conducted on *E. coli* O157:H7 treated with the same method used in the transcriptome analysis. Six DEGs (fatty acid biosynthesis, *accB*; ribosome, *rpsS*; oxidative phosphorylation, *frdA* and *atpG*; amino acid metabolism, *argG* and *ansB*) from different pathways were selected to confirm the RNA-seq results using the primers listed in Appendix A. The annealing temperature was 56 °C. Relative gene expression was calculated using the 2^−ΔΔCT^ method with *16S rRNA* as an internal control.

### 2.10. Statistical Analysis

All experiments were performed in triplicate. Data were expressed as the mean ± standard deviation. The differences between groups were analyzed using one-way analysis of variance, followed by Duncan’s tests, using SPSS software (version 23; SPSS, Inc., Chicago, IL, USA); *p* ≤ 0.05 was considered statistically significant in the physiological experiments.

## 3. Results

### 3.1. Determination of Sublethal Injury and Repair Conditions

Following previous studies, the SI *E. coli* O157:H7 survived on the NS medium but not on the SC medium, while the average *E. coli* O157: H7 survived on both NS and SC (when the content of NaCl in SC was ≤3%) [22,23]. To obtain the SI model with an SI rate of more than 99.99% [24], *E. coli* O157:H7 cells were treated by HHP at various pressures (100, 200, 300, 400, and 500 MPa) and times (1, 3, and 5 min). As shown in Figure 1A (Appendix A), the SI rate increased with increasing pressure. When the pressures reached 400 MPa (5 min) and 500 MPa (1, 3, and 5 min), the SI rate exceeded 99.99%, and the intact cell content was 0.00%. Similar results were reported by Somolonos et al. [25], in which the sub-lethal injury rate reached 99.99% when treated with 400 MPa for 5 min. The SI cell content negatively correlated with pressure (Appendix A). When the pressure reached 500 MPa, the amounts of sub-lethal cells were lower than 0.05%, which was too low to carry out the subsequent transcriptome analysis. Consequently, samples treated with 400 MPa and 5 min (HHP group) were used in the subsequent analysis.

The repair of SI *E. coli* O157:H7 cells in TSB, BPW, PBS, and minA is shown in Figure 1B. As demonstrated by the consistent growth trend of survival counts in NS and SC, the SI cells could be recovered over time. When repaired in TSB and BPW, the remaining SI cells were completely repaired after 2 h of incubation. In PBS and minA, the repair of SI cells needed 5 h and >5 h, respectively, suggesting a positive correlation between the repair efficiency and nutrient abundance. Similarly, Bi et al. [22] found that the trend of SI *E. coli* repair efficiency induced by high-pressure carbon dioxide was TSB > carrot juice > peptone water > PBS, which is positively related to nutrient abundance. Thus, the SI *E. coli* O157: H7 repaired in TSB for 2 h was selected for further investigation of the recovery mechanism of sub-lethal injured *E. coli* O157:H7 using transcriptome analysis.

### 3.2. Physiological Analysis Reveals the Repair Mechanism

#### 3.2.1. Determination of the Intracellular Protein and ATP Contents

The intracellular protein contents of UT, HHP, and repair groups are shown in Figure 2A. Compared with the UT group, the intracellular protein content decreased to 0.34 mg/mL after HHP treatment, which could be due to the leakage of intracellular protein [26]. During the repair process, the protein content increased slightly from 4 h; however, the content remained significantly lower than the HHP group. The intracellular ATP contents of treatments are exhibited in Figure 2B, which showed no significant change when treated with a single HHP. However, it started to increase rapidly after four hours of repair. When repaired for 8 h, the ATP content was about seven times higher than that of the UT group. In summary, the contents of protein and ATP in cells showed an upward trend after the repair.

#### 3.2.2. Observation of Morphological Changes

The morphological changes were observed using an AFM, which has been used to investigate biosystems, such as bacteria and eukaryotic, because of its ability to reveal structural details with unprecedented resolution. Figure 2C shows that the UT cells demonstrated short rod shapes with continuous and smooth surfaces. When treated with HHP, the uniform short rod shape was broken into fragments (red arrow). There were hollows on the cell surface (green arrows), suggesting the destruction of cell morphology. In the 4 h repair group, the unrepaired hollows on the cell surface remained (green arrows). After 8 h of repair, the cells returned to full rod shape and began proliferating. In total, 20 locations (509.6 nm × 509.6 nm) were randomly selected to calculate the roughness (Figure 2D). The HHP group showed the highest roughness, while the repair groups were lower than the HHP group but significantly higher than the UT group. Following a previous study, the morphology of SI *E. coli* O157:H7 returned to normal levels during repair, as seen under a scanning electron microscope (SEM) [27], similar to our results. These findings suggest that the morphology recovers during the repair process, which requires more time.

#### 3.2.3. Determination of the Enzyme Activities

Figure 2E shows the enzyme activities of two crucial members in the peroxisome (SOD and CAT). The SOD activity decreased significantly after sub-lethal treatment induced by HHP. After an 8 h repair process, even though the SOD activity showed an upward trend, its activity (881.35 U/mgprot) remained significantly lower than the UT group (3641.61 U/mgprot). The CAT activity increased more than twice that of the UT group (3.17 U/mgprot) after HHP treatment. During repair, the CAT activity showed an increasing trend. When repaired for 6 h, the CAT activity recovered to the same level as the UT group. When repair continued for another two hours, the CAT activity reached 4.04 U/mgprot. Inaoka et al. [28] found that the disruption of the gene encoding SOD reduced the viability of HHP-treated *B. subtilis*; however, the disruption of the gene encoding CAT had no detectable effect on the viability of HHP-treated cells. The SOD and CAT activities showed opposite trends in our study, possibly due to the gene encoding SOD being more sensitive than CAT in resisting the oxidative stress caused by HHP. HHP treatment might result in a state of metabolic imbalance, which, in its turn, causes a burst of reactive oxygen species (ROS) [29]. However, oxidative defense mechanisms could be induced during the cell repair process to avoid ROS accumulation [30], which could be responsible for increasing SOD and CAT activities.

The tendencies of Na^+^K^+^-, Ca^2+^Mg^2+^-, and total ATPase activities are shown in Figure 2F. These ATPases were significantly inhibited after sub-lethal injury induced by HHP. During the repair process, these ATPase activities continued to decrease until 6 h and increased after 8 h of repair. Similarly, Ma et al. [27] found that the Na^+^K^+^-ATPase and Ca^2+^Mg^2+^-ATPase activities did not increase until 12 h of repair. This discrepancy could be because SI cells are repaired in PBS rather than TSB, which is nutrient-deficient. These findings suggest that increased SOD, CAT, and ATPases activities reflect physiological injuries, and the repair effect might be positively correlated with nutrient abundance.

### 3.3. Changes in Transcript Levels of E. coli O157:H7 during HHP and Repair 

#### 3.3.1. Transcriptional Response to the HHP and Repair 

A total of 1,067,811,900, 1,093,588,200, and 922,627,500 raw reads were collected from UT, HHP, and repair groups using the Illumina Hiseq platform (Appendix A). After filtration, 1,003,731,547, 986,057,925, and 861,150,652 clean reads were collected. Of these clean reads, 6,367,292 (UT), 6,063,864 (HHP) and 5,784,618 (Repair) reads were mapped to the reference genome, and the mapping ratios were all higher than 84% (Appendix A); i.e., these data were of high quality for further analysis. The points representing the same treatment were aggregative based on principal component analysis (PCA) (Figure 3A). Those of UT, HHP, and repair treatments were distributed in different quadrants, suggesting that the results of the same treatments were repeatable, and there were significant differences among different treatments. A total of 1304, 1071, and 1154 DEGs were detected in the UT vs. HHP, HHP vs. repair, and UT vs. repair groups, respectively (Figure 3B). In the HHP vs. repair group, there were more upregulated DEGs (767) than downregulated DEGs (304). By contrast, the downregulated DEGs in the UT vs. HHP and UT vs. repair groups (1081 and 818) were higher than that of upregulated DEGs (223 and 336). It could be speculated that the HHP process inhibited the expression of DEGs while the repair process advanced. Venn analysis was applied to identify the similarities and differences among DEGs for the UT vs. HHP, HHP vs. repair, and UT vs. repair groups (Figure 3C,D). There were 51 DEGs in common among the three groups: 16 upregulated and 35 downregulated (Appendix A). These DEGs were related to responses to environmental changes, such as acid (*hycBCDEFG* and *citCDEFXG*) [31,32,33] and nitrogen starvation (*yeaGH* and *astABCDE*) [34,35]. These results suggest that the changes in the internal and external environment induced by the HHP and repair process could lead to responses at the transcriptional level.

To validate the RNA-Seq data, a qRT-PCR assay was used to determine the expression of DEGs with the same RNA samples. For all six DEGs tested (*accB*, *rpsS*, *frdA*, *atpG*, *argG,* and *ansB*), the qRT-PCR results followed RNA-Seq results (Figure 3E), suggesting the validity of the RNA-Seq results.

#### 3.3.2. Cluster Analysis, GO, and KEGG Analysis

Considering that genes with similar expression patterns are likely to possess similar functions or participate in the same regulatory pathways, all 1942 DEGs in the UT vs. HHP and HHP vs. repair were clustered using the short time-series expression miner (STEM, Figure 4A). Based on the cluster analysis, the DEGs could be clustered into eight profiles, in which profiles 1 and 2 were significantly enriched (*p*-value ≤ 0.05, Figure 4B). Profiles 1 and 2 presented the unresponsive and upregulated DEGs, respectively, which were downregulated following HHP. A total of 1168 DEGs were contained in these two profiles, in which 437 genes were enriched in profile 1, and 731 genes were enriched in profile 2. Then, the DEGs contained in profiles 1 and 2 were subjected to GO term analysis and KEGG pathway enrichment analysis (Appendix A and Figure 4C). In GO analysis, the DEGs were classified into three categories, molecular function (MF), biological process (BP), and cellular component (CC). As shown in Figure 4C (Appendix A), under the CC category, the most abundant sub-categories of profile 1 were: membrane (GO: 0016020), membrane part (GO: 0044425), and intrinsic component of membrane (GO: 0031224). For profile 2, the sub-categories binding (MF), metabolic process (BP), and ribosome (CC) were significantly enriched. For KEGG pathway analysis, the most abundant pathways are shown in Appendix A. The ‘C5-branched dibasic acid metabolism (ko00660)’, ‘Cell cycle-Caulobacter (ko04112)’ and ‘Valine, leucine, and isoleucine biosynthesis (ko00290)’ were the top three pathways enriched in profile 1. For profile 2, ‘Ribosome (ko03010)’, ‘Alanine, aspartate, and glutamate metabolism (ko00250)’, and ‘Aminoacyl-tRNA biosynthesis (ko00970)’ were the top enriched groups. These findings suggest that DEGs responding to the HHP and repair processes are associated with the membrane, genetic information transmission, and energy biosynthesis.

### 3.4. Response of DEGs Related to Membrane after HHP and Repair Processes

DEGs responding to HHP and repair treatments identified in the membrane-related pathways included ‘Peptidoglycan biosynthesis’ and ‘Fatty acid biosynthesis,’ which were associated with membrane biosynthesis (Figure 5A,B, Appendix A). Peptidoglycan is a component of the cell wall (also called outer membrane) in bacteria [36]. In total, 12 DEGs were involved in the ‘Peptidoglycan biosynthesis’ pathway, in which 5 DEGs were promoted, and 7 showed no significant change during the repair process (Appendix A). Deleting the genes *murA* and *murB* directly decreased peptidoglycan synthesis in *Corynebacterium glutamicum* [37]. It could be inferred that the upregulation of these genes might lead to repair the outer membrane.

Conversely, gene *dacC* (encoding the low-molecular-weight penicillin-binding protein necessary for synthesis and stabilization) showed no significant change. Gao et al. [38] found that the downregulation of gene *murD* in *Staphylococcus aureus* resulted in a thinner cell wall. The unresponsiveness of these DEGs might reduce the repair efficiency of the outer membrane. 

Fatty acids in *E. coli* are produced for the biosynthesis of lipids and the inner membrane. Most DEGs involved in the ‘Fatty acid biosynthesis’ pathway were significantly induced after repair. The *fabB* gene (encoding beta-ketoacyl-ACP synthase I, which is required for unsaturated fatty acid biosynthesis [39]) increased by 1.43 log_2_FC compared to the HHP group. The *fadG* gene (an essential gene in this pathway) increased by 0.91 log_2_FC. It was worth noting that the *fadD* gene showed an opposite trend, downregulated 0.62 log_2_FC after the repair process. The product of *fadD* participates in the initial stages of long-chain fatty acid β-oxidation, which oxidizes fatty acid into acetyl-CoA [40]. It could be inferred that promoting fatty acid synthesis-related genes and inhibiting its oxidation-related gene should be a method of how the inner membrane was repaired at the transcriptional level.

### 3.5. Response of DEG Related to Energy Biosynthesis after HHP and Repair 

‘Oxidative phosphorylation’ is the pathway that produces ATP, the energy currency in bacteria. There are five complexes in this pathway (Figure 6A, Appendix A), including NADH dehydrogenase (complex I), succinate dehydrogenase, cytochrome bc1 complex, cytochrome c oxidase, and ATP synthase (complex V). The *nuo* genes encode the proton pump, which catalyzes the first step of electron transport [41]. Six of the nine *nuo* genes upregulated 1.04 to 2.25 log_2_FC after repair, except *nuoA*, *nuoH*, *and nuoI*. F_0_F_1_-ATP synthase (complex V) is responsible for final ATP synthesis by oxidative or photophosphorylation in membranes [42]. As shown in Figure 6, the genes *atpADG* (encoding the F_0_F_1_ ATP synthase subunits α, β, and γ, respectively) and *atpC* (encoding the F_1_ complex subunit ε) were all upregulated during the repair. It could be inferred that the upregulation of genes involved in the oxidative phosphorylation pathway led to an increase in energy synthesis, thereby meeting the needs of the repair process.

### 3.6. Response of DEG Related to Genetic Information Transmission after HHP and Repair 

DEGs involved in the genetic information transmission in response to HHP and repair treatment were identified, enriched in ’aminoacyl-tRNA biosynthesis‘ and ’ribosome’. The DEGs involved in these pathways showed an identical trend of change (i.e., they were promoted after repair treatment; Figure 6B, Appendix A). Concerning the expressions of genes involved in ‘aminoacyl-tRNA biosynthesis’, 19 of the 22 DEGs are amino acid ligases, which activate the amino acid and transfer this moiety to tRNA [43]. The ribosome is where the codon translates into amino acid and is related to the polypeptide elongation. The products of DEGs enriched in ‘ribosome’ are in EF-Tu (*rpmC*, *rplBCDPVW*, and *rpsCJS*), EF-G (*rpsGL*), IF1 (initiation factor 1) and IF 3 (*rpsDKM* and *rpmI*), SecY (*rplFORX*, *rpmD*, and *rpsEHNQ*), and FtsY (*rpsAP* and *rplYS*), which covers the entire process from the binding of tRNA and mRNA to the transport of proteins; all these genes were promoted. In particular, the *rpmE2-1* gene, whose product is involved in the essential release factor RF1, increased 16.20 log_2_FC after repair. In prokaryotes, the native function of RF1 is to recognize codon UAA/UAG and terminate translation [44]. The relationship between HHP and RF1 is unclear; however, it might be a critical point for responding to HHP injuries and assisting in repair. The upregulation of these DEGs suggests enhancing the genetic information transmission process.

## 4. Discussion

The SI *E. coli* O157:H7 model (treated with 400 MPa for 5 min) was repaired when cultured for 2 h in TSB. Then, the repair mechanism was studied using physiological and transcriptome analysis.

The intracellular protein content increased slightly after 4 h of repair, similar to Pan et al. [45], who found that the protein content of sub-lethal *L. monocytogenes* induced by Ar/O_2_ plasma only slightly increased after the repair process. The ribosome is the site of protein biosynthesis. Ribosome reconstruction is crucial for the recovery of growth in HHP-injured *B. subtilis* [46]. Similarly, the DEGs involved in ribosome were significantly upregulated in our study (Figure 6B), which could be the reason for the promotion of protein content. However, the protein content did not increase substantially. Leucine plays a vital role in regulating the signal transduction of translation initiation [47]. As shown in Appendix A, the ‘valine, leucine, and isoleucine biosynthesis’ pathway was enriched in profile 1. The essential *leuABC* operon involved in this pathway (which encodes enzymes catalyzing the conversation of α-ketoisovalerate to leucine) showed no significant change [48,49]. This phenomenon could be a reason for the insignificant increase in protein content. These findings suggest that the promotion of protein content could be due to the upregulation of the ribosome pathway; however, the process of protein synthesis is affected by complex factors leading to inefficient protein synthesis.

Conversely, the content of ATP increased significantly during repair (Figure 2B). The mechanism of the ATP content increase is unclear; however, a significantly positive correlation between ATP content and viability was detected [50], suggesting better cell viability after repair treatment. Gao et al. [51] found that the mitochondria complex I subunit was upregulated in the radioresistant glioma U87MG cells, and the copy numbers of mitochondria increased. Gurgan et al. [52] found that the downregulation of *nuo* genes and *atpCD* impair energy production of *Rhodobacter capsulatus* by heat stress. Similar trends of these DEGs were detected in our transcriptome analysis (Figure 6A and Appendix A). The ATP content, that substantially increased during the repair process, is due to the promotion of the ‘oxidative phosphorylation’ pathway. Repairing the SI cells, especially the cytoplasmic membrane, is energy-dependent and requires RNA and protein synthesis [53,54]. It could be speculated that the cell repair process is accompanied by the protein and ATP synthesis, which could be responses to physiological activity demands.

In terms of cell morphology, it appears that cells were repaired; however, the roughness was only partially recovered (Figure 2C). Nikparvar et al. [55] observed that membrane holes in *L. monocytogenes* were not repaired until 48 h, when they were treated with 400 MPa for 8 min using a quantitative model. Thus, it could be concluded that the cell membrane was not completely repaired after 8 h of repair. Combined with the transcriptomic results (Figure 5A,B), we speculate that the repair of the outer membrane was partially inhibited, and that of the inner membrane was promoted. Arroyo et al. [56] found that the membrane structure of pulsed electric field induced SI in *Enterobacter sakazakii* (also a Gram-negative organism) could be repaired with different kinetics. The repair of the outer membrane was much slower than the plasma membrane at the initial stages, shown indirectly by our results. In other words, even though the number of colonies was restored, the physiological function of cells remained in the repair process after culture in TSB for 2 h. In terms of the membrane part, the repair of the inner membrane could occur before that of the outer membrane, which led to failure in restoring the roughness.

## 5. Conclusions

The repair mechanism of HHP-induced SI *E. coli* O157:H7 includes the following: repair of the inner membrane of cells by promoting fatty acid biosynthesis and then restoring cell morphology; generation of energy by promoting oxidative phosphorylation process; and accelerating protein synthesis by promoting ribosome function. Our findings suggest that the inner membrane is an essential repair site, and the repair process requires energy and protein. These findings provide insight into the repair process of HHP-induced SI *E. coli* O157:H7, which could help avoid potential food safety hazards caused by SI cells

## Figures and Tables

**Figure 1 foods-11-02377-f001:**
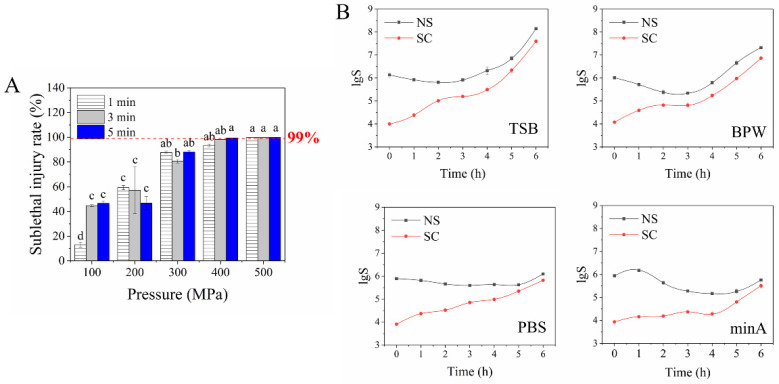
The sub-lethal injury rate (**A**) and the repair of sub-lethal *E. coli* O157: H7 in TSB, BPW, PBS, and minA mediums (**B**). Letter a–d in (**A**) indicate statistically significant within the groups treated with different pressures and same time (*p* ≤ 0.05, Duncan test).

**Figure 2 foods-11-02377-f002:**
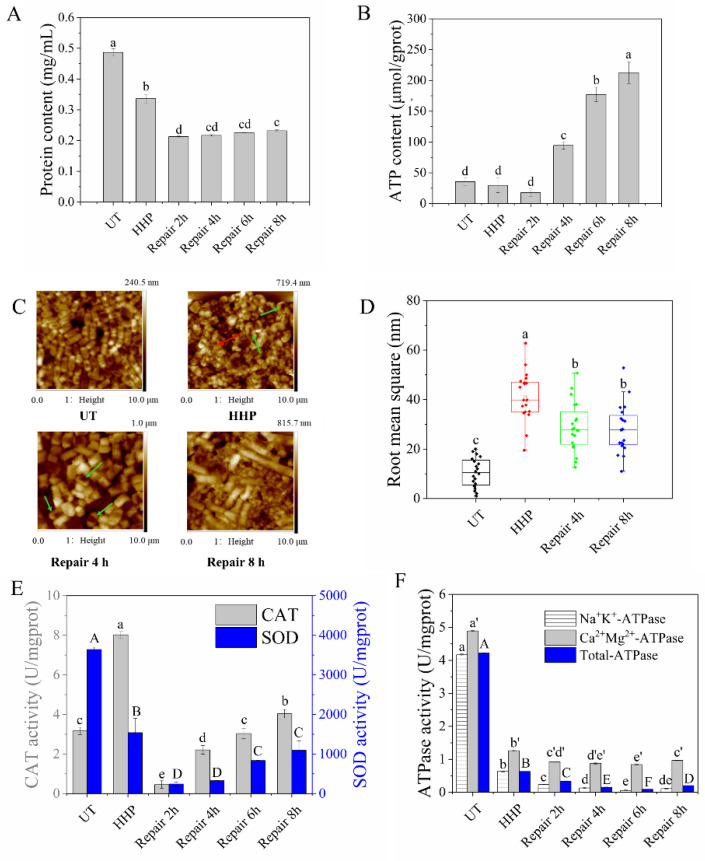
Effects of different repair times on intracellular protein content (**A**), intracellular ATP content (**B**), AFM images (**C**), surface roughness (**D**), CAT and SOD activities (**E**), and ATPase activities (**F**). (**C**) contained the 2-dimensional and 3-dimensional AFM images of UT, HHP, and repair groups (repaired for 4 h and 8 h). The red arrow represented the cell fragments and the green arrows holes. The average bacterial surface roughness in nm was obtained on 509.6 × 509.6 nm^2^ area of UT, HHP and repair groups. For each group, 20 cells were plotted randomly. Different letters (a to e, A to F, and a’ to e’) indicate statistically significance between different groups (*p* ≤ 0.05, Duncan test).

**Figure 3 foods-11-02377-f003:**
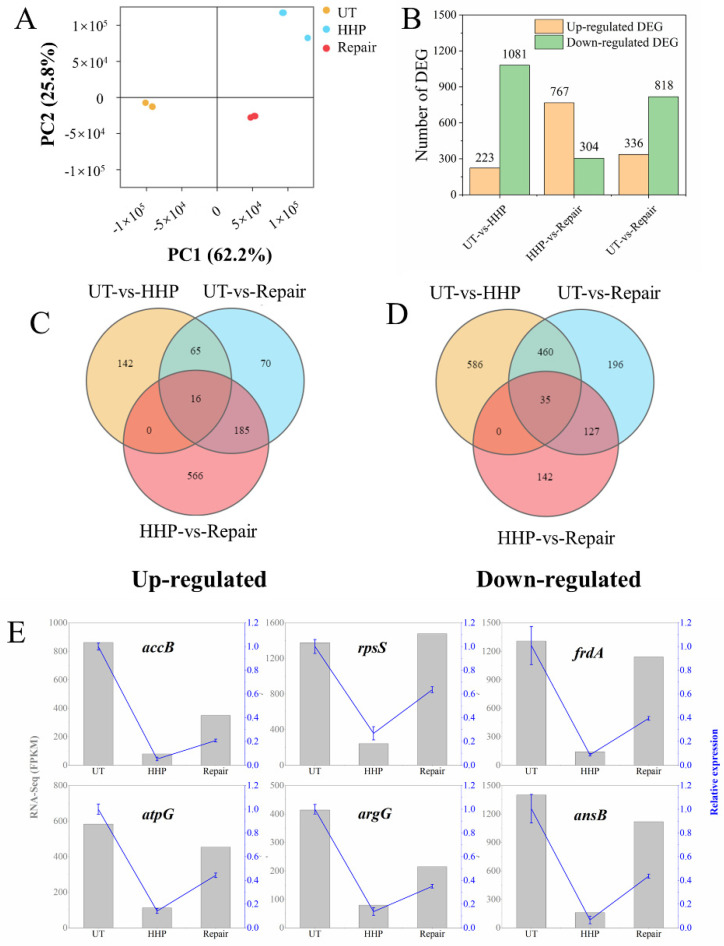
Analysis of the *E. coli* O157: H7 transcriptome in HHP (sub-lethal cells) and repair (repaired in liquid medium for 2 h) groups. (**A**) the principal component analysis of UT, HHP, and repair groups. (**B**) The upregulated and downregulated differentially expressed genes (DEGs) among UT vs. HHP, HHP vs. repair, and UT vs. repair. (**C**,**D**) The Venn diagram analysis of upregulated and downregulated DEGs enriched in both the HHP and repair groups, respectively. (**E**) The qRT-PCR results of the 6 DEGs, which were the mean values of 2^−ΔΔCT^ obtained from three biological replicates with error bars representing standard deviations. Results were normalized using *16S rRNA* and expressed as fold change.

**Figure 4 foods-11-02377-f004:**
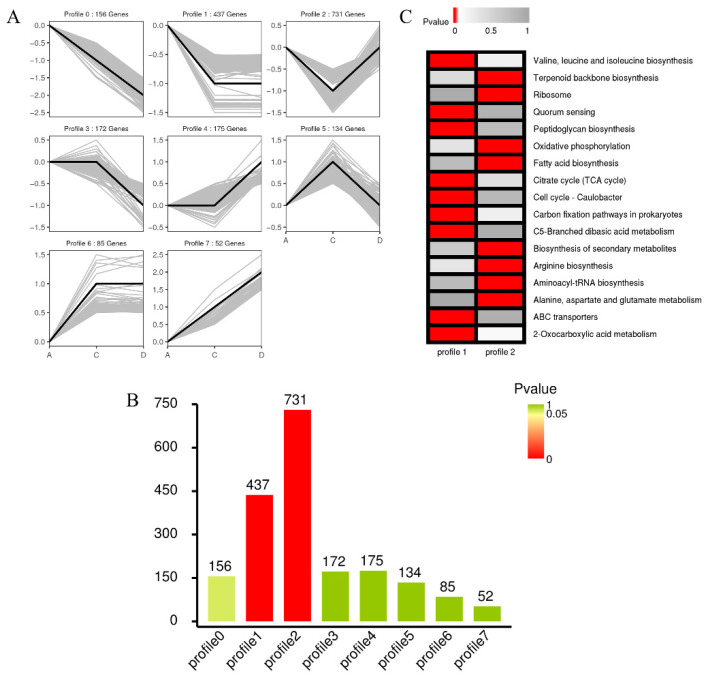
Cluster analysis of the *E. coli* O157: H7 transcriptome in HHP (sub-lethal cells) and repair (repaired in liquid medium for 2 h) groups. (**A**) Patterns of gene expressions across UT, HHP, and repair groups inferred by STEM analysis. In each profile, the light grey lines represented the expression pattern of each gene enriched, while the black line represented the expression tendency of all the genes in this profile. (**B**) The amounts of DEGs and significances in each profile, in which red columns defined significant difference. (**C**) KEGG enrichment analysis of profile 1 and profile 2. The significance of the most enriched pathway in these two clusters was indicated by the *p*-value. The red regions represented the significant *p*-values, whereas the grey regions represented the non-significant values.

**Figure 5 foods-11-02377-f005:**
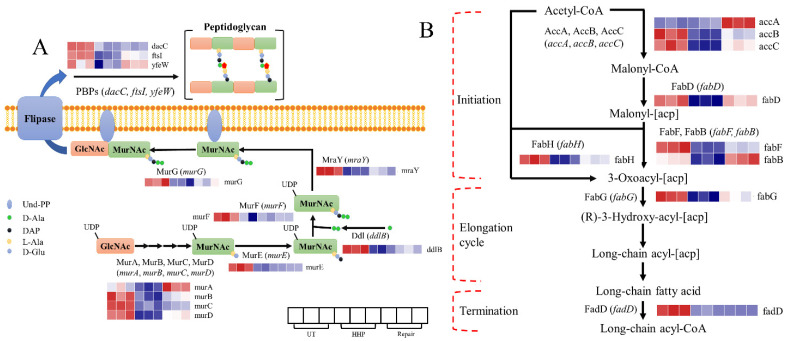
Transcriptional changes of genes involved in the significantly enriched pathway. (**A**) The peptidoglycan biosynthesis pathway. (**B**) The fatty acid biosynthesis pathway.

**Figure 6 foods-11-02377-f006:**
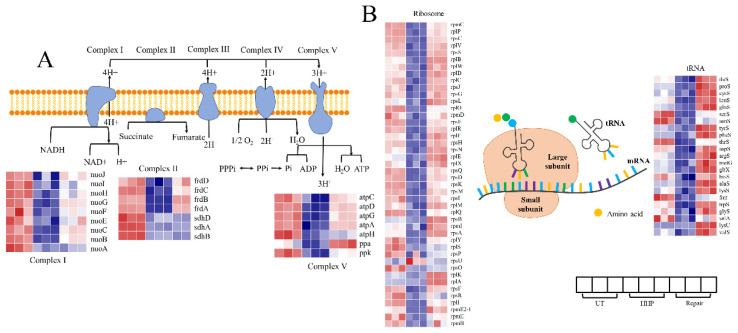
Transcriptional changes of genes involved in the significantly enriched pathway. (**A**) The oxidative phosphorylation pathway. (**B**) The ribosome and aminoacyl-tRNA biosynthesis pathways.

## Data Availability

Data is contained within the article and Appendix A.

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
