# Peer review of "Integrative Physiological and Transcriptome Analysis Reveals the Mechanism for the Repair of Sub-Lethally Injured Escherichia coli O157:H7 Induced by High Hydrostatic Pressure"

_foods, 2022, doi:10.3390/foods11152377_

Round 1
Reviewer 1 Report
Dear authors
Thanks for this interesting study
First of all, I encourage you to review the english to improve the grammar and style, for example
line 32> owing to --> because of
line 39> and recognized --> it is recognized
line 59> lof SI --> log of SI
line 450> which leads to 450 the lag of protein synthesis ¿?, what means?
Also, to try to avoid the repeat to nexus such as, meantime, nowadays so on or phrasal verbs, and even didn´t
The third parragraph is absolutly neccessary to improve.
Material and methods
line 85> how many measurements the authors made?, please specify the exact OD not approx.
Please, expand for another exacted measurements.
Why the authors have used SPSS software, usually this software is rather for clinical studies than "scientific", actually there are plenty of platforms for transcriptomic studies, e.g, MAX-QUANT/PERSEUS (which is really easy to use).
What happened if the p-value is fixed to p <0.01 or adjusted?. Usually the cut-of 0.01 is more severe and better (from my viewpoint) or at least the Benjamin-Hochberg adjustmet is more "realistic". So, could the author repeat the statistical analysis according to that? Thanks
Results:
Fig 1. Could merge all figures B in one?, exapnd for other taht it could be possible, for example fig3. In this way is easier to compare.
Please, to rewrite the legends
Figure 4 is hard to intreprete the results, so one solution is to make it bigger or to move to supp data. The same for fig 5
Discussion
Please, make the conclusion!
Thanks
Author Response
Dear reviewer,
The responses to your comments have been finished. Please see the attachment.
Yours sincerely
Jing-Yi Hao

Reviewer 2 Report
This study explores the repair mechanisms of HHP induced sublethally injured Escherichia coli O157:H7. The study is well presented and organized, and the only limitation is the fact that the authors test only one strain of E. coli and hence they can explore differences or similarities between strains. Why didn’t the authors test more strains? Was it because of the volume of the transcriptome analysis? The results show that the intracellular protein content, ATP content and enzyme activities (SOD, CAT, and ATPase) increased, and the morphology repaired. Overall the work adds information to the scientific literature.
Author Response

(The authors gave the same response as above.)

Reviewer 3 Report
This manuscript entitled "Integrative physiological and transcriptome analysis reveals the mechanism for the repair of sublethally injured Escherichia coli O157:H7 induced by high hydrostatic pressure" was mainly focused on the repair mechanisms of HHP induced SI Escherichia coli O157:H7. The experiment had been designed well and the results obtained had also been written, interpreted and discussed very well. In my point of view, this manuscript is nice, easy to be read, and coped with the state-of-the-art applied studies interested in the food safety.
1. L5: check the punctuations
2. In keywords: the word “repair” is a bit vague, please replace
3. L48; please replace “Escherichia coli” by “E. coli” and the whole manuscript for this issue.
4. L93: please change “minutes” to “min” , you should follow the standard unit abbreviations.
5. The figure's text is not clear and difficult to read, please use a higher resolution image with text.
6. Data Availability Statement is not supported
7. Generally, the manuscript requires linguistic revision
Author Response

(The authors gave the same response as above.)

Round 2
Reviewer 1 Report
Dear author,
Even if the manuscript is interesting, there are 2 main limitations linked to the statistical analysis, the platform used to, and the non-adjustment.
So, I could acept to use SPSS to the t-test, however the adjustment must be done, even if you do not find statistical significances. And hereby, the reason why I suggested you to use, for example MAX-QUANT platform (free of charge), because you can describe the fold change or enrichment.
So, plese re-write the manuscript according to the new results(benjamin-Hochberg adjustem), and at least comment the limitations.
Thanks
